

# Virus like particles as a platform for cancer vaccine development

Hui Kian Ong[1], Wen Siang Tan[2,3] and Kok Lian Ho[1]

[1] Department of Pathology, Faculty of Medicine and Health Sciences, Universiti Putra Malaysia, Serdang, Selangor, Malaysia
[2] Department of Microbiology, Faculty of Biotechnology and Biomolecular Sciences, Universiti Putra Malaysia, Serdang, Selangor, Malaysia
[3] Institute of Bioscience, Universiti Putra Malaysia, Serdang, Selangor, Malaysia

## ABSTRACT

Cancers have killed millions of people in human history and are still posing a serious health problem worldwide. Therefore, there is an urgent need for developing preventive and therapeutic cancer vaccines. Among various cancer vaccine development platforms, virus-like particles (VLPs) offer several advantages. VLPs are multimeric nanostructures with morphology resembling that of native viruses and are mainly composed of surface structural proteins of viruses but are devoid of viral genetic materials rendering them neither infective nor replicative. In addition, they can be engineered to display multiple, highly ordered heterologous epitopes or peptides in order to optimize the antigenicity and immunogenicity of the displayed entities. Like native viruses, specific epitopes displayed on VLPs can be taken up, processed, and presented by antigen-presenting cells to elicit potent specific humoral and cell-mediated immune responses. Several studies also indicated that VLPs could overcome the immunosuppressive state of the tumor microenvironment and break self-tolerance to elicit strong cytotoxic lymphocyte activity, which is crucial for both virus clearance and destruction of cancerous cells. Collectively, these unique characteristics of VLPs make them optimal cancer vaccine candidates. This review discusses current progress in the development of VLP-based cancer vaccines and some potential drawbacks of VLPs in cancer vaccine development. Extracellular vesicles with close resembling to viral particles are also discussed and compared with VLPs as a platform in cancer vaccine developments.

## INTRODUCTION

Vaccination remains the most effective approach in the prevention and control of infectious diseases. Eradication of fatal smallpox virus in 1979 represented an extraordinary milestone in vaccinology (*Strassburg, 1982*). Presently, the majority of commercial vaccines are formulated to fight infectious diseases, but cancer vaccines are rarely reported. Nevertheless, the idea of prospective cancer vaccination was suggested by Dr. William Coley who "vaccinated" cancer patients intratumorally with inactivated *Streptococcus pyogenes* and *Serratia marcescens*, which became known as Coley's Toxin in 1891 (*McCarthy, 2006*). The rationale behind his attempt was prompted by his observation of sarcoma remission

Corresponding author
Kok Lian Ho, klho@upm.edu.my

in patients who had developed erysipelas. Coley's Toxin was claimed to be an effective immunotherapeutic agent against cancer but unfortunately, the treatment was viewed as a scientific controversy at that time (*Guo et al., 2013*). Today, a better understanding of tumor immunology has proven the efficacy of Coley's Toxin. Moreover, improved strategies of vaccine design have allowed the development of many potential cancer vaccines over the past 60 years (*Schlom, 2012*). However, the process of translation of experimental cancer vaccines into effective therapeutic agents for clinical use is often challenging. To date, licensed cancer vaccines are only available for liver, cervical, and prostate cancers (*Cuzick, 2015*; *Graff & Chamberlain, 2015*; *Kushnir, Streatfield & Yusibov, 2012*; *Ma, Roden & Wu, 2010*).

According to the National Cancer Institute of the United States, cancer vaccines are biological response modifiers that stimulate or restore the function of the immune system to fight infections and diseases (*Lollini et al., 2006*). In general, cancer vaccines are categorized into (i) preventive cancer vaccines and (ii) therapeutic cancer vaccines (*Liu, 2014*). The former preparations, also known as prophylactic vaccines, are effective against oncoviruses, such as hepatitis B virus (HBV), hepatitis C virus (HCV), human papillomavirus (HPV), Epstein-Barr virus (EBV), Kaposi's sarcoma-associated herpesvirus, human T-cell lymphotropic virus, and merkel cell polyomavirus (*Parkin, 2006*). Table 1 summarizes information about these oncoviruses and diseases associated with them. By definition, administration of a preventive cancer vaccine elicits immune responses of the host against the invasion of oncoviruses, whereas cancer therapeutic vaccines are a form of immunotherapy that stimulates and enhances patient's own immune system to fight against pre-existing cancers (*Liu, 2014*). Compared to the current conventional cancer treatments, which include invasive surgeries, chemotherapy, and radiotherapy, immunotherapy approach is less invasive, has fewer detrimental side effects, and can confer long-term cancer remission or even cancer immunity (*Dimberu & Leonhardt, 2011*).

Tumor-induced immunosuppression in tumor microenvironment (TME) is one of the primary factors that impede the development of cancer vaccines (*Wang et al., 2017*). Several immunosuppressive leukocytes, including myeloid derived suppressor cells (MDSCs), tumor associated macrophages (TAMs) and regulatory T cells (Tregs) resident at the TME sites, release immunosuppressive cytokines, such as TGF-β and IL-10, which promote tumor growth, metastasis, and angiogenesis (*Cesana et al., 2006*). TAMs also inhibit T-cell activation and trigger apoptosis of activated T-cells, which greatly reduces the amount of tumor infiltrating lymphocytes (*Kuang et al., 2009*; *Williams, Yeh & Soloff, 2016*). In addition, poorly immunogenic tumor self-antigen also exhibits immunological tolerance that leads to ineffective counteracting immune responses (*Guo et al., 2013*). Therefore, an effective cancer vaccine must be capable of eliciting strong antitumor immune responses to overcome TME immunosuppressive state and break self-tolerance.

Successful vaccination relies heavily on the immunogenicity and presentation efficiency of the tumor antigen. Various entities have been employed to function as tumor antigen delivery platforms. In this regard, virus-like particles (VLPs) offer several advantages over other substances. VLPs are multimeric nanostructures morphologically resembling authentic viral particles. They are mainly composed of viral structural proteins with

**Table 1  Oncoviruses and associated cancers.**

| Virus | Oncoviruses associated cancers | Percentage of the cancers caused (%) | Mechanism of Carcinogenesis | Reference |
|---|---|---|---|---|
| HBV | Hepatocellular carcinoma | 3.1 | Chronic inflammation | *De Martel & Franceschi (2009)* and *Parkin (2006)* |
| HCV | Hepatocellular carcinoma | 1.8 | Chronic inflammation | *De Martel & Franceschi (2009)* and *Parkin (2006)* |
| HPV | Cervical, vulva, vagina, penis, anus and oropharynx cancers. | 5.2 | Direct carcinogens act via expression of viral oncoproteins | *Parkin (2006)* and *Yim & Park (2005)* |
| EBV | Burkitt's lymphoma, Hodgkin lymphoma, B cell lymphoma, nasopharyngeal carcinoma, gastric and sporadic carcinoma. | 1–2 | Direct carcinogens act via expression of viral oncoproteins | *Hsieh et al. (1996)* and *Parkin (2006)* |
| HTLV-1 | Adult T-cell leukemia | 0.3 | Direct carcinogens act via expression of viral oncoproteins | *Jeang (2010)* and *Parkin (2006)* |
| KSHV | Kaposi's sarcoma and primary effusion lymphoma | 1 | Direct carcinogens act via expression of viral oncoproteins | *Parkin (2006)* and *zur Hausen (2001)* |
| MCV | Merkel cell carcinoma | NA | Direct carcinogens act via expression of viral oncoproteins | *Parkin (2006)* |

inherent self-assembly properties but are devoid of viral genetic materials (*Chroboczek, Szurgot & Szolajska, 2014*). Thus, VLPs are non-replicative and non-infectious, which greatly enhances their safety (*Noad & Roy, 2003*).

In general, VLPs can be classified into non-enveloped and enveloped VLPs. The formers are self-assembled from at least one viral protein which is expressed using a suitable host expression system such as mammalian cells, insect cells, yeasts, bacteria and cell free systems without acquiring any host component (*Kushnir, Streatfield & Yusibov, 2012*). Yeasts and mammalian cells are the most commonly employed expression systems in the production of commercialized HBV and HPV VLP-based vaccines (*Kushnir, Streatfield & Yusibov, 2012*). However, production of more structurally complex non-enveloped chimeric VLPs involves the display of heterologous epitopes or peptides on the surface of VLPs via genetic engineering (Fig. 1A) (*Yong et al., 2015a*; *Yong et al., 2015b*). Non-enveloped VLPs can also be chemically conjugated with a target antigen via hetero-bifunctional chemical linker such as sulfosuccinimidyl 4-(N-maleimidomethyl)cyclohexane-1-carboxylate (sulfo-SMCC) and nanoglue, in which, the chimeric VLPs can be produced without extensive genetic alteration and this overcomes the limitation imposed on VLP formation (Fig. 1B) (*Biabanikhankahdani et al., 2016*; *Jemon et al., 2013*; *Lee et al., 2012*). On the other hand, enveloped VLPs acquire part of the host cell membranes as their lipid envelope, where a foreign epitope or peptide could be integrated and displayed on the surface (*Gheysen et al., 1989*; *Haynes, 2009*). Similar to non-enveloped VLPs, enveloped VLPs can be produced by expressing several viral structural proteins in a suitable expression system. Alternatively, a

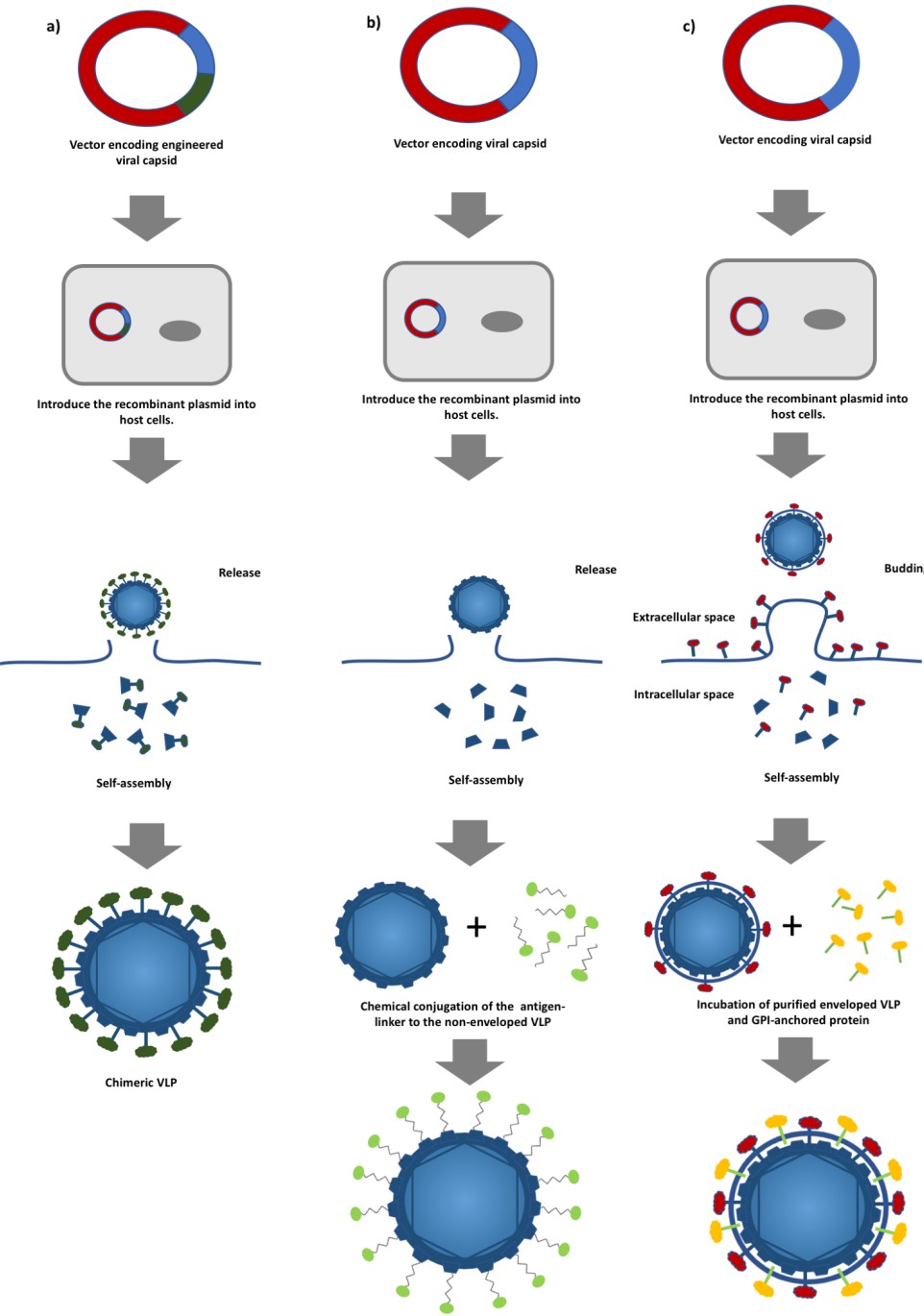

**Figure 1** **A schematic diagram of virus-like particles (VLPs) production using different approaches.**
(A) Production of non-enveloped chimeric VLPs using genetic alteration. Viral structural protein is
fused with a foreign antigen via genetic engineering followed by the expression of the chimeric protein
in a suitable host system. (B) Production of non-enveloped chimeric VLPs by chemical conjugation.
Non-enveloped VLPs are produced by expressing a viral structural protein, and surface decoration of the
VLPs is achieved by conjugating a foreign antigen to the VLPs, and (C) production of enveloped chimeric
VLPs by the protein transfer approach. Enveloped VLPs are produced by expressing the viral proteins in
suitable host cells followed by incubating with the glycosylphosphatidylinositol (GPI)-anchored proteins.
The foreign antigens are then transferred to the lipid bilayer of the VLPs.

protein transfer technique can be employed to display the heterologous epitopes or peptides on the surface of the enveloped VLPs. This approach allows spontaneous incorporation of the glycosylphosphatidylinositol (GPI)-anchored protein or other immunostimulatory molecules to the lipid bilayer of the enveloped VLPs via a simple incubation step (Fig. 1C) (*Patel et al., 2015a*). The primary advantage of the protein transfer technology in the production of chimeric enveloped VLP is its ability to retain the functionality of the incorporated protein without an extensive genetic modification (*Shashidharamurthy et al., 2012*). In addition, attributed to the viral origin of VLPs, some of the VLP-based vaccines are self-adjuvating, in which, they contain the pathogen associated molecular pattern (PAMP) of viruses that could potentially enhance the activation of innate immune systems via the Toll-like receptors and pattern recognition receptors (*Crisci, Barcena & Montoya, 2012*; *Rynda-Apple, Patterson & Douglas, 2014*). These self-assembling, engineerable, and safe VLPs can be leveraged to display various tumor antigens for targeting different cancers (*Patel et al., 2015a*; *Patel et al., 2015b*). Genetic fusion of a tumor antigen to a non-self-antigen, such as a virus coat protein, was previously reported to notably improve its immunogenicity (*Savelyeva et al., 2001*). Tumor self-antigens incorporated onto the surface of VLPs can be efficiently taken up, processed, and presented by specialized dendritic cells (DCs). Figure 2 shows that priming of $CD8^+$ cytotoxic T lymphocytes (CTLs) and $CD4^+$ T helper cells by activated DCs elicits potent antitumor immune responses and breaks self-tolerance (*Barth et al., 2005*; *Guo et al., 2013*; *Li et al., 2013*). Furthermore, VLPs displaying a tumor antigen (VLP-DTA) do not contain auto-antigens, thus they offer an alternative scaffold with a much lower risk of autoimmunity in relative to the whole tumor antigen vaccines which present the entire spectrum of potential tumor antigens, including auto-antigen to DC (*Schirrmacher & Fournier, 2010*). Several studies also indicated that VLPs could overcome the immunosuppressive state of the TME characterized by a significant decrease in immunosuppressive leukocyte population and levels of corresponding cytokines in immunized animals (*Cubas et al., 2011*; *Li et al., 2008*).

The ultimate objective of a cancer vaccine is to produce cell-mediated immune responses, specifically Th1 immune responses, for efficient activation of $CD8^+$ T-cells or CTLs, which are critical for both the clearance of infected cells and destruction of cancerous cells (*Maecker et al., 1998*). The capability of VLP-DTA to selectively elicit strong cell-mediated and humoral immunities against tumor cells while reducing the risk of autoimmunity makes it an ideal candidate for cancer vaccine delivery platform (*Jennings & Bachmann, 2008*). This review discusses current progress in the development of VLP-based cancer vaccines against (i) hepatocellular carcinoma, (ii) cervical cancer, (iii) pancreatic cancer, (iv) prostate cancer, (v) breast cancer, (vi) skin cancer, vii) lung cancer, and (viii) oncovirus-associated cancers (see Table 2).

## SURVEY METHODOLOGY

In this study, we reviewed articles related to VLP-based cancer vaccines. All references in this review paper were retrieved using search engines such as PubMed, Scopus, Google Scholar and ResearchGate. Keywords such as Virus-like particles, cancer vaccines, cytotoxic lymphocyte, tumor antigen and oncovirus were used to search for the references.

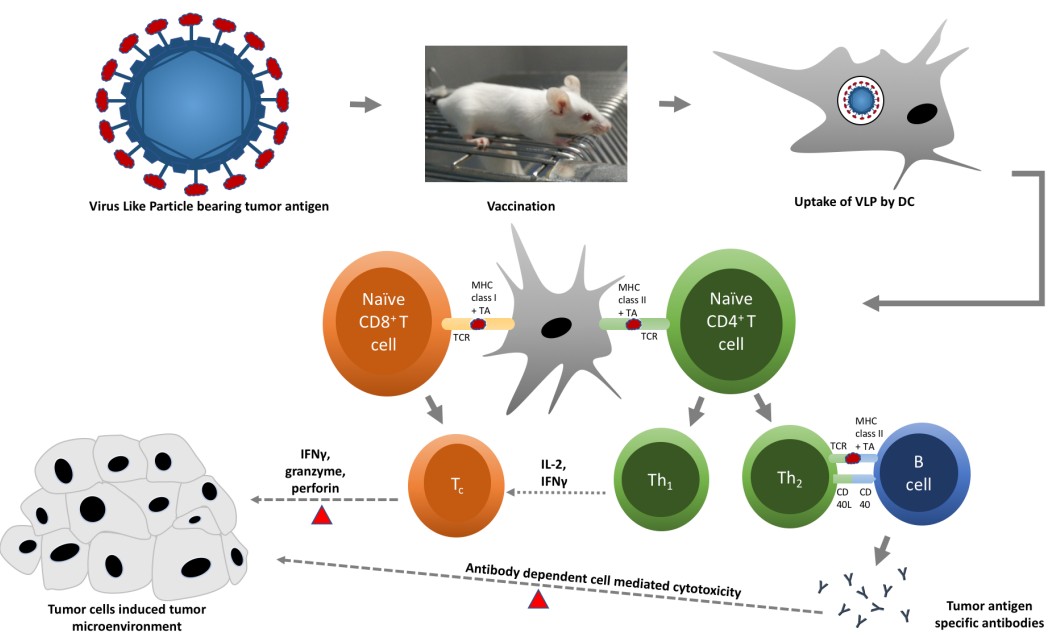

**Figure 2** **A schematic diagram of virus-like particle (VLP) in triggering specific immune responses against tumor cells.** Chimeric VLPs displaying tumor antigen (red oval) on their surface are administered into an animal model. Tumor antigen is taken up, processed, and presented by dendritic cells (DCs) to respective T cells. Presentation of tumor antigen by DCs converts naïve cytotoxic and helper T cells into cytotoxic ($T_c$) and effector helper (Th) T cells, respectively. $T_c$ cells (red triangles) kill tumor cells by releasing cytotoxic proteins, such as granzyme and perforin. $Th_1$ cells support the activation of $T_c$ cells by releasing interleukin-2 (IL-2) and interferon gamma ($IFN\gamma$), whereas $Th_2$ cells stimulate B cells to produce tumor antigen-specific antibodies, which are capable to bind and mark (red triangles) tumor cells for effective killing by natural killer (NK) cells and phagocytes.

## Hepatitis B- and hepatitis C-associated hepatocellular carcinoma

Primary liver cancers are ranked as the third leading cause of cancer-related mortality in the world (*Altekruse et al., 2014*). The 5-year relative survival rate of liver cancer patients is estimated to be 25–45% (*Kim et al., 2014*). Approximately one third of the world population are infected with HBV, whereas about 350 million people worldwide developed chronic infection (*Lavanchy, 2005*). In the United States, around 50% of liver cancers are caused by HBV and HCV (*Ly et al., 2012*). Despite the presence of effective HBV preventive vaccines, low accessibility to such vaccines in resource-limited countries and the lack of therapeutic vaccines have retarded the eradication of HBV. Liver cancer caused by HCV infection, however, is expected to increase in the following decades unless preventive HCV vaccines become universally available. To effectively reduce the incidence of oncovirus-associated liver cancers, vaccinations with cancer preventive vaccines against HBV and HCV are essential, and continuing development of therapeutic vaccines is required to treat patients with pre-existing infection.

Since the invention of the first licensed HBV vaccine, tremendous amount of effort has been applied to the development of optimized HBV vaccines. To date, there are 12 VLP-based HBV vaccines on the market. Typical examples include Engerix-B, Enivac HB, Gene

**Table 2** Summary of the VLP based cancer vaccines.

| No. | VLPs | Cancer antigen targeted | Viral antigen targeted | Types of vaccines | Reference |
|---|---|---|---|---|---|
| **Viral Like Particles and Hepatocellular Carcinoma** | | | | | |
| 1 | Hamster polyomavirus | – | HBsAg | Therapeutic | *Pleckaityte et al. (2015)* |
| 2 | *Macrobrachium rosenbergii* nodavirus | – | 'a' determinant | Preventive | *Yong et al. (2015a)* |
| 3 | HBsAg | – | HVR1 | Preventive | *Vietheer et al. (2007)* |
| 4 | HCV | – | E1 and E2 | Preventive | *Elmowalid et al. (2007)* |
| 5 | Retrovirus | – | E1 and E2 | Preventive | *Garrone et al. (2011)* |
| 6 | HBc | MAGE-1 | – | Therapeutic | *Zhang et al. (2007)* |
| 7 | HBc | MAGE-3 | – | Therapeutic | *Zhang et al. (2007)* |
| 8 | HBc | AFP1 | – | Therapeutic | *Zhang et al. (2007)* |
| 9 | HBc | HBx | – | Therapeutic | *Ding et al. (2009)* |
| **Virus Like Particles and HPV Associated Cervical Cancer** | | | | | |
| 1 | HPV | | L1 and L2 | Preventive | *Huber et al. (2015)* and *Pineo et al. (2013)* |
| 2 | IBDV | E7 | | Therapeutic | *Martin Caballero et al. (2012)* |
| 3 | RHDV | E6 | | Therapeutic | *Jemon et al. (2013)* |
| **Virus Like Particles and Pancreatic Cancer** | | | | | |
| 1 | SIV | Trop2 | | Therapeutic | *Cubas et al. (2011)* |
| 2 | SHIV | hMSLN | | Therapeutic | *Li et al. (2008)* |
| 3 | SHIV | mMSLN | | Therapeutic | *Zhang et al. (2013)* |
| **Virus Like Particles and Prostate Cancer** | | | | | |
| 1 | Murine polyomavirus | PSA | | Preventive | *Eriksson et al. (2011)* |
| **Virus Like Particles and Breast Cancer** | | | | | |
| 1 | Murine polyomavirus | Her2 | | Therapeutic and preventive | *Tegerstedt et al. (2007)* and *Tegerstedt et al. (2005)* |
| 2 | Influenza A virus | Her2 | | Preventive | *Patel et al. (2015b)* |
| **Virus Like Particles and Skin Cancer** | | | | | |
| 1 | HBc | MAGE-3 | | Therapeutic | *Kazaks et al. (2008)* |
| 2 | Murine polyomavirus | OVA (model antigen) | | Therapeutic | *Brinkman et al. (2005)* |
| 3 | Murine polyomavirus | TRP2 | | Therapeutic | *Brinkman et al. (2005)* |
| 4 | RHDV | gp33 (model antigen) | | Preventive | *McKee et al. (2012)* |
| **Virus Like Particles and Lung Cancer** | | | | | |
| 1 | HBc | CLDN18.2 | | Preventive | *Klamp et al. (2011)* |
| **Virus Like Particles and EBV Associated Cancers** | | | | | |
| 1 | NDV | | gp350/220 | Preventive | *Ogembo et al. (2015)* |
| 2 | EBV | | | Preventive | *Ruiss et al. (2011)* |

Vac-B, Hepavax-Gene, and Recombivax HB. All commercial VLP-based HBV vaccines are produced by expressing hepatitis B virus surface antigen (HBsAg) in yeasts or mammalian expression systems (*Kushnir, Streatfield & Yusibov, 2012*). In an attempt to improve efficacy of current vaccines, *Pleckaityte et al. (2015)* innovatively incorporated heterologous HBsAg specific single-chain fragment variable into VLPs of hamster polyomavirus. *In vitro* virus neutralization assay demonstrated promising antiviral activity of the vaccine, but its protective efficacy has yet to be investigated *in vivo*. This vaccine is believed to be a potential therapeutic HBV vaccine which could provide curative effects to chronic HBV patients (*Pleckaityte et al., 2015*). Recently, *Macrobrachium rosenbergii* nodavirus (*Mr*NV) VLPs were employed to display an HBV antigenic determinant known as the "*a*" determinant (*Yong et al., 2015a*), an immunodominant region responsible for the induction of HBV-specific humoral immune responses (*Howard & Allison, 1995*; *Ruiz-Tachiquin et al., 2007*). Immunization of mice with *Mr*NV chimeric VLPs induced the production of "*a*" determinant-specific antibodies and, surprisingly, the induced antibody levels were significantly higher than those observed after the administration of the commercial HBV vaccine Engerix-B, which served as a positive control in that study (*Yong et al., 2015a*).

With respect to HCV, it has been reported that approximately 3% of the world population are infected with HCV. About 75% of patients acutely infected with HCV developed a chronic liver disease, of which, 3–11% developed liver cirrhosis, which may eventually lead to hepatocellular carcinoma (HCC) (*Alter, 2006*; *Micallef, Kaldor & Dore, 2006*; *Poynard et al., 2003*; *Somi et al., 2014*). In the development of VLP-based vaccines against HCV infection, HBsAg-derived VLPs have been used to display the heterologous antigenic epitope hypervariable region 1 (HVR1) of HCV (*Vietheer et al., 2007*). Following the vaccination of mice with chimeric HBV VLPs expressing HVR1, humoral immune response was manifested as the production of HVR1 specific antisera in immunized animals. Moreover, *in vitro* studies also demonstrated the protective effects of the vaccine (*Vietheer et al., 2007*). Surprisingly, the antigenicity and immunogenicity of HBsAg were preserved even after the insertion of the HVR1 peptide. This finding potentially paves way to the development of multivalent hepatitis vaccines.

HCV VLPs are commonly synthesized by expressing viral glycoproteins E1 and E2. The efficacy of HCV VLP vaccine was extensively tested in different animal models, including mouse and baboon, and yielded satisfactory results (*Jeong et al., 2004*; *Lechmann et al., 2001*; *Murata et al., 2003*; *Qiao et al., 2003*). Recently, chimpanzees, the only HCV-susceptible animal model, were used to evaluate the protective efficacy of HCV VLPs. These immunization studies suggested that HCV VLPs were capable of conferring protection upon the vaccinated animals against HCV challenge by eliciting E1- and E2-specific humoral and cell-mediated immune responses (*Elmowalid et al., 2007*). Similarly, in a different study, enveloped retroviral VLPs pseudotyped with heterologous HCV E1 and E2 glycoproteins also elicited strong cross-reactivity in both mouse and macaque (*Garrone et al., 2011*). It was reported that E1- and E2-specific neutralizing antibody responses might be the major effectors against HCV infection (*Beaumont & Roingeard, 2013*; *Pestka et al., 2007*).

Non-virus induced HCC cannot be treated with preventive vaccine, but a VLP-based antitumor vaccine has been previously described by *Zhang et al. (2007)*, who successfully incorporated tumor associated antigens (TAAs) of HCC, which include melanoma associated antigen 1 (MAGE-1), melanoma associated antigen 3 (MAGE-3), and α-fetoprotein (AFP1), into HBV core antigen (HBc) derived VLPs. Similarly, to the native viruses, these chimeric HBc VLPs could be presented by DCs. Immunization with DCs pulsed with chimeric HBc VLPs also activated strong CTL immune responses that correlated with decreased mortality of immunized animals. Pulsing DCs with VLPs was expected to improve the danger signal and DC presentation, and thereby enhance immunization efficiency (*Zhang et al., 2007*).

Recently, chimeric HBc VLPs bearing the HBV x protein (HBx) were produced by *Ding et al. (2009)*. The HBx gene is commonly integrated into hepatocyte DNA and plays a pivotal role in tumorigenesis of HBV-associated HCC (*Arbuthnot, Capovilla & Kew, 2000*; *Schaefer et al., 1998*; *Ye et al., 2008*). HBV infected hepatocytes and associated HCC were demonstrated to express this viral antigen (*Su et al., 1998*). Vaccination of mice with HBc VLPs expressing HBx either in the form of active immunization or by an adoptive transfer has resulted in specific cell-mediated immune responses (*Ding et al., 2009*). Most of the VLP-based vaccines designed against HBV-associated HCC are preventive and aimed to eliminate HBV viruses by targeting a specific viral antigen but intriguingly, this therapeutic vaccine was designed to target the viral antigen expressed by infected hepatocytes.

## Human papilloma virus-associated cervical cancer

HPV is the main causative agent of cervical cancer (CC). According to the American Cancer Society, the survival rate of CC patients varies between 15% and 93%, depending on cancer stage. It was reported that nearly 100% of women with CC were found to be HPV positive, of which, HPV-16 and HPV-18 accounted for approximately 70% of all CC cases (*Nour, 2009*). To date, the most effective way of CC prevention is early vaccination against HPV. Multivalent Gardasil, Cervarix, and Gardasil 9, which mainly target HPV-16 and HPV-18, are the only licensed preventive VLP-based HPV vaccines available on the market. These commercially available HPV vaccines were developed based on HPV VLPs that express L1, the major capsid protein of HPV (*Kushnir, Streatfield & Yusibov, 2012*). Despite the success of HPV vaccination programs, these preparations are type-restricted and, therefore, do not target less prevalent but high-risk HPV subtypes that cause about 30% of CC cases (*Tjalma et al., 2013*). Furthermore, the high cost of vaccine production has also prompted scientists to develop a broader spectrum second generation HPV vaccine. It is generally believed that the highly conserved L2 minor capsid protein of HPV is an ideal candidate for second generation HPV vaccines (*Tyler, Tumban & Chackerian, 2014*). L2 is one of the structural proteins produced during the later stage of HPV replication (*Graham, 2010*). Vaccination with L2 peptide elicited cross-neutralizing antibodies and conferred cross-protection upon immunized animals, although the induced antibody titer was significantly lower than that stimulated by HPV L1-based VLP (*Gambhira et al., 2007*; *Pastrana et al., 2005*). Therefore, *Pineo, Hitzeroth & Rybicki (2013)* created a new vaccine by incorporating L2 into the C-terminal region of L1 in an attempt to improve its immunogenicity. The

fusion peptide L1/L2 was expressed in *Nicotiana benthamiana* and L2 was found to be exposed on the surface of L1/L2 chimeric VLPs. Immunization studies demonstrated that L1/L2 chimeric VLPs induced both L1- and L2-specific humoral immune responses and conferred cross-protection upon immunized mice against HPV-16 and HPV-52 (*Pineo, Hitzeroth & Rybicki, 2013*). Similar studies were carried out by *Huber et al. (2015),* in which a cross-neutralizing epitope of the L2 protein of HPV-45 was incorporated into HPV-18 L1 protein. The resulting chimeric VLP vaccine 18L1-45RG1 was observed to cross-protect against HPV-18, HPV-39, HPV-45, and HPV-68 when mice were passively immunized with antisera from the 18L1-45RG1 immunized rabbits (*Huber et al., 2015*). Remarkably, a much boarder spectrum L2-expressing HPV VLP vaccine was recently reported to be capable of cross-neutralizing 24 HPV subtypes *in vitro*, while conferring cross-protection upon the immunized animals against 21 HPV subtypes (*Schellenbacher et al., 2013*).

To date, all commercially available CC vaccines are prophylactic HPV vaccines. The lack of a therapeutic vaccine makes it impossible to cure pre-existing HPV-associated CC. Nevertheless, an attempt to develop a therapeutic vaccine against HPV was undertaken by inserting the non-structural E7 protein of HPV into infectious bursal disease virus VLPs (VLP E7) (*Martin Caballero et al., 2012*). The E7 protein is a tumor-specific antigen (TSA), which is highly expressed in HPV-associated CC. It is suggested to be an important element in the maintenance of transformed phenotype of cancerous cells (*Baker et al., 1987*; *DeFilippis et al., 2003*; *Psyrri et al., 2004*). Tumor challenge followed by vaccination of mice with VLP E7 demonstrated a complete rejection of the tumor (*Martin Caballero et al., 2012*).

To improve CTL activity, an MHC class II restricted T-cell epitope was inserted into VLPs. In particular, *Jemon et al. (2013)* incorporated the pan HLA DR-binding epitope (PADRE), a potent T-cell inducer, into rabbit hemorrhagic disease virus VLPs expressing the E6 protein (RHDV VLP). Similar to E7, E6 is a TSA expressed in HPV-associated CC that plays a major role in malignant conversion (*Yim & Park, 2005*). The combination of E6-RHDV-VLP-PADRE vaccine with antibody treatments reactivated T cells, reduced TC-1 tumor outgrowth, and increased the survival rate of immunized animals (*Jemon et al., 2013*). Apart from CC, HPV was also reported to cause cancers of the vulva, vagina, penis, anus, as well as oropharynx cancers (*Ljubojevic & Skerlev, 2014*). Therefore, HPV vaccines have potential to confer a certain degree of protection against these HPV-associated diseases.

## Pancreatic cancer

Pancreatic cancer is a highly aggressive cancer, which is currently ranked as one of the top five causes of cancer-related mortality (*Gostimir et al., 2016*). The average survival rate of pancreatic cancer patients from the onset of diagnosis is approximately 20% and decreases dramatically to 6% within 5 years (*Sannino et al., 2016*). The extremely low 5-year relative survival rate of pancreatic cancer patients is largely attributed to the lack of disease-specific symptoms and early cancer cell dissemination (*Oberstein & Olive, 2013*; *Rhim et al., 2012*). Current treatments of pancreatic cancer rely heavily on conventional therapies, such as surgery, chemotherapy, and radiotherapy.

Trop2 is one of the tumor-associated antigens (TAAs), which is overexpressed in many epithelial carcinomas. Overexpression of this protein is often associated with tumorigenesis, metastasis, decreased overall survival rate, and tumor grade (*Cubas et al., 2009*; *Fong et al., 2008*). Limited expression of Trop2 by healthy cells makes it a promising TAA for cancer immunotherapy targeting (*Cubas et al., 2009*). Recently, Trop2 was incorporated into enveloped simian immunodeficiency virus VLPs in order to generate chimeric mTrop2 VLPs (*Cubas et al., 2011*). Immunization of mice with mTrop2 VLPs alone following a tumor challenge significantly reduced tumor size and increased mouse life expectancy by approximately 36% (*Cubas et al., 2011*). The higher survival rate (70%) was achieved when mice immunized with mTrop2 VLPs were also treated with gemcitabine (*Cubas et al., 2011*). Immunization with mTrop2 VLPs overcame self-tolerance and elicited both innate and cell-mediated immunities manifested by the elevation of levels of antigen-specific tumor-infiltrating T lymphocytes, natural killer (NK) cells, and NK T-cells. In contrast, the numbers of immunosuppressive MDSCs and Tregs as well as the levels of immunosuppressive cytokines were notably reduced by this immunization. Furthermore, expression levels of IL-2, IL-13, and IFN-$\gamma$ were enhanced by mTrop2 VLP administration, demonstrating Th1-skewed immune responses that are necessary for the destruction of cancer cells. In addition, mTrop2-specific antibodies were detected in the serum and demonstrated to play a critical role in mediating cell cytotoxicity against cancerous cells (*Cubas et al., 2011*).

Intriguingly, a recombinant simian/human immunodeficiency virus (SHIV) also has been recruited as backbone for the production of VLP-based pancreatic cancer vaccine. The first SHIV VLPs were produced by a baculovirus expression system using the protein transfer technique (*Yao et al., 2000*). Following the development of SHIV VLPs, *Li et al. (2008)* incorporated the recently identified pancreatic cancer TAA mesothelin (MSLN) into SHIV VLPs. MSLN is commonly overexpressed in pancreatic cancer cells, and it has been shown to cause metastasis in ovarian cancer cells when it interacts with mucin 16 (*Gubbels et al., 2006*). Due to its low expression level in healthy cells, MSLN was recently listed as a potential biomarker for early detection of pancreatic cancer (*Laheru & Jaffee, 2005*). Previous studies showed that mice immunized with chimeric SHIV VLPs expressing human MSLN (hMSLN VLP) were protected from tumor challenge and their life expectancy increased to ∼60 % (*Li et al., 2008*). Immunization with hMSLN VLPs of mice grafted with human pancreatic cancer cells elicited both cellular and humoral immunities, in particular, strong CTL activity, which resulted in a significant reduction of tumor mass. High population of IFN-$\gamma$-secreting T lymphocytes also indicated Th1-skewed immune responses. The numbers of immunosuppressive lymphocytes, including different subsets of Tregs, decreased significantly upon immunization, which correlated with prolonged survival of immunized mice (*Li et al., 2008*). As hMSLN and mouse MSLN (mMSLN) only share about 55% homology in amino acid sequence, strong immune responses elicited in a previous study (*Li et al., 2008*) could be attributed to xenogeneic immune responses against hMSLN (*Wang et al., 2009*). To investigate the capability of the vaccine to break self-tolerance, *Zhang et al. (2013)* continued the study by substituting hMSLN with mMSLN to generate chimeric, mMSLN-expressing VLPs (mMSLN VLPs).

Surprisingly, immunization with mMSLN VLPs followed by a tumor challenge in mice has successfully broken self-tolerance and elicited strong mMSLN-specific CTL activity that significantly reduced tumor mass and increased the overall survival rate of the animals. Notable reduction in the frequency of Tregs was also believed to be one of the major drivers of cancer growth inhibition (*Zhang et al., 2013*).

In addition to its association with pancreatic cancer, MSLN is also commonly expressed in mesothelioma, ovarian cancer, lung cancer, acute myeloid leukemia, and uterine serous carcinoma (*Chang & Pastan, 1996*; *Frierson Jr et al., 2003*; *Ho et al., 2007*; *Steinbach et al., 2007*). As a common TAA in different cancer cells, the capability of MSLN-expressing VLPs to confer cross-protection against different tumor challenges is an interesting area to explore.

## Prostate cancer

Prostate cancer is one of the leading causes of cancer-related mortality among men in the United States and Western Europe and is ranked as the third most common cancer in Europe (*Ferlay et al., 2013*). Early stage localized prostate cancer is often treated with prostatectomy, hormonal therapy, and radiotherapy (*Wilt & Ahmed, 2013*). However, almost all patients with advanced prostate cancer tend to experience bone metastasis that reduces the effectiveness of early therapies (*Msaouel et al., 2008*). Chymotrypsin-like serine proteases or prostate specific antigens (PSAs) are prostatic secretory proteins that commonly serve as biomarkers for the detection of prostate cancer because they are highly expressed by prostate cancer cells (*Koie et al., 2015*). These androgen-regulated antigens are expressed in extremely low levels in non-prostate healthy tissues and in normal prostate tissues. Therefore, they are appropriate tumor antigens for cancer targeting as prostate tumors overexpress PSA (*Cunha et al., 2006*). To date, the only licensed therapeutic prostate cancer vaccine available on the market is Sipuleucel-T, a cell-based preparation that involves leukapheresis and *in vitro* DC activation (*Graff & Chamberlain, 2015*).

In VLP-based prostate cancer vaccine development, chimeric murine polyomavirus VLPs harboring full length PSA (PSA-MPy VLPs) conferred partial protection from prostate tumor challenge in mice and the maximum protection was achieved only when PSA-MPy VLPs were loaded onto DCs in the presence of CpG as an adjuvant (*Eriksson et al., 2011*). Both CTL activity and CD4$^+$ immune responses were observed following the immunization but not the humoral immune responses because PSA antigen was speculated to be buried inside the viral capsid, and therefore, remained inaccessible by B cells (*Eriksson et al., 2011*). Loading the VLPs into DCs is expected to activate the latter and enhance the danger signal. Nevertheless, immunotherapy using PSA as a tumor antigen will most likely result in prostate destruction due to autoimmune reactions directed against PSA, which is also expressed on normal prostate tissues.

## Breast cancer

Breast cancer is a notorious fatal illness that primarily affects women. Its 5-year relative survival rate is the highest at the early stage (nearly 100%) and drops to 22% at stage IV or metastatic stage (*Torre et al., 2015*). The most common treatment of breast cancer

includes invasive mastectomy that may severely impact body self-esteem (*Markopoulos et al., 2009*). A VLP-based breast cancer vaccine has been developed by incorporating the breast cancer TAA HER-2/neu (Her2) into murine polyomavirus VLPs to generate chimeric Her2$_{1-683}$Py VLPs (*Tegerstedt et al., 2005*). Her2 has been commonly used as a biomarker for early detection of breast cancer, and its overexpression is often associated with tumor growth, increased mortality, and cancer relapse rate (*Ross et al., 2003*). *In vivo* immunization studies followed by tumor challenge indicated that a single vaccination with Her2$_{1-683}$Py VLPs was sufficient to elicit a strong Her2-specific cellular immunity that led to tumor rejection and longer survival period. Immunization with Her2$_{1-683}$Py VLPs of transgenic mice expressing mutated Her2 oncogene also resulted in complete protection against tumor outgrowth, demonstrating preventive efficacy of the vaccine (*Tegerstedt et al., 2005*). However, immunization timing seems to have major consequences on vaccine efficacy as delayed immunization only postponed tumor outgrowth without conferring any significant protection (*Tegerstedt et al., 2005*). Humoral immunity was not detected in that study as Her2 was probably expressed internally within the VLPs. As previously mentioned, loading VLPs into DCs may enhance DC activation. In a more recent study by the same group, *Tegerstedt et al. (2007)* loaded Her2$_{1-683}$Py VLPs into murine DCs and immunized mice with such activated DCs. The results indicated that immunization efficiency was improved and a relatively smaller amount of Her2$_{1-683}$Py VLPs was required to confer protection against tumor challenge if VLPs were loaded into DCs (*Tegerstedt et al., 2007*). Polyomavirus VLP-based vaccine preparations have an advantage over other VLPs due to the fact that numerous polyomavirus receptors are expressed on different cells, including sentinel cells (*Drake et al., 2000*).

Furthermore, enveloped VLPs also have been used to display Her2. *Patel et al. (2015b)* modified Her2 TAA to a glycosylphosphatidylinositol (GPI)-anchored form and incorporated the fusion GPI-HER2 protein into enveloped influenza VLPs. The newly created vaccine (GPI-HER2-VLP) was shown to be immunogenic and capable of activating both cellular and humoral immune responses. The survival rate of mice immunized with GPI-HER2-VLP increased to approximately 66%. Intriguingly, the Her2-specific antibody level was found to be similar in mice immunized with GPI-HER2-VLP and GPI-HER2. However, the latter treatment failed to confer significant protection against tumor challenge, indicating that the cellular immunity must play a major role. This correlated well with Th1-skewed immune response observed in the study (*Patel et al., 2015b*). Her2 is a novel TAA, which is also overexpressed in many different cancers, such as endometrial carcinoma, gastric cancer, salivary duct carcinoma, lungs adenocarcinoma, and ovary cancer (*Chiosea et al., 2015*; *Meza-Junco, Au & Sawyer, 2011*; *Ruschoff et al., 2012*; *Santin et al., 2008*). However, apart from the case of breast cancer, the potential of Her2-bearing VLPs as vaccine candidates against other types of cancer has yet to be investigated.

## Skin cancer

Skin cancer is generally divided into melanomas and non-melanomas, and it has been reported to be one of the most common cancers in the world (*Simoes, Sousa & Pais, 2015*). White population is usually more susceptible to skin cancer than other races (*Narayanan,*

*Saladi & Fox, 2010*). To date, there is no available vaccine against skin cancers, but a bacteriophage Qβ VLP-based melanoma cancer vaccine (MelQbG10) developed by Cytos Biotechnology AG has now completed its Phase 2 clinical trial and is expected to be the first melanoma cancer vaccine on the market (*Goldinger et al., 2012*; *Kushnir, Streatfield & Yusibov, 2012*). Other VLPs, including those based on extensively studied HBc capsid, were recently utilized in VLP-based melanoma vaccine development as demonstrated by *Kazaks et al. (2008)*. In this vaccine, melanoma associated antigen 3, a TAA whose overexpression is often associated with poor prognosis and melanoma metastasis, was incorporated into HBc VLPs. To further improve the immunogenicity of the vaccine, single-stranded CpG oligonucleotides were packaged into the VLPs. Although the immunogenicity of this vaccine has yet to be investigated *in vivo*, it is expected that it will be a suitable vaccine candidate to elicit strong cellular immunity (*Kazaks et al., 2008*).

In another report, *Brinkman et al. (2005)* incorporated H-2K$^b$-restricted ovalbumin (OVA)$_{257-264}$ epitope and H-2K$^b$-restricted CTL epitope of melanoma differentiation antigen tyrosinase-related protein 2 (TRP2) into VP$_1$ of murine polyomavirus VLPs (PLP), generating chimeric PLP (VP$_1$-OVA$_{252-270}$ PLP) and chimeric murine polyomavirus like pentamers (VP$_1$-TRP2$_{180-192}$PP), respectively. *In vivo* studies of VP$_1$-OVA$_{252-270}$ PLP in mice showed that notable protection against OVA-expressing melanoma cells was conferred by the induction of strong CTL activity. Although VP$_1$-TRP2$_{180-192}$PP failed to assemble into VLPs, surprisingly it partially protected immunized animals from lethal melanoma challenge (*Brinkman et al., 2005*). These PLP-based vaccines demonstrated highly efficient antigen carriers for inducing CTL responses, underlining their potential as immunotherapeutic against cancer.

In addition, rabbit hemorrhagic disease virus (RHDV) VLPs have been recruited as tumor antigen delivery system because this virus does not infect humans (*Steinmetz et al., 2010*). RHDV VLPs have an advantage over human-infecting viruses, such as HBV or polyomavirus, as pre-existing immunities of the vaccinated individual may interfere with successful treatments. RHDV VLPs were engineered to express the model antigen gp33 and conjugated with α-galactosylceramide that acted as an immunostimulatory adjuvant. Vaccination of mice followed by tumor challenge with B16 melanoma expressing gp33 activated intense CTL and invariant NKT activity manifested by significant elevations of IL-4 and IFN-γ levels (*McKee et al., 2012*; *Tomura et al., 1999*).

## Other cancers
### Lung cancers
A VLP-based vaccine against lung metastases has been established by incorporating the TAA isoform 2 of the tight junction molecule claudin-18 (CLDN18.2) into HBc VLPs (*Klamp et al., 2011*). Immunization of mice with this vaccine followed by a tumor challenge with CLDN18.2-expressing CT26 colon cancer cells resulted in lower tumor burden around the lung areas (*Klamp et al., 2011*). Protection conferred by this VLP-based vaccine was largely mediated by complement-dependent cytotoxicity and antibody-dependent mediated cytotoxicity. Surprisingly, antibodies elicited by this vaccine were highly specific against CLDN18.2 and not cross-reactive with CLDN18.1 variant, which is normally

expressed on healthy lung tissue, despite high homology of their protein sequences (*Klamp et al., 2011*).

### Epstein-Barr virus related cancers

Burkitt's lymphoma, Hodgkin lymphoma, B cell lymphoma, nasopharyngeal carcinoma, gastric and sporadic carcinoma may develop in Epstein-Barr virus (EBV) infected individuals (*Hjalgrim, Friborg & Melbye, 2007*; *Thompson & Kurzrock, 2004*). In fact, EBV infects more than 90% of the worldwide population, but most people remain asymptomatic. However, cancer development may be triggered in EBV-infected individuals by numerous factors (*Hjalgrim, Friborg & Melbye, 2007*). Recently, chimeric Newcastle disease virus (NDV) VLPs containing heterologous antigenic glycoprotein gp350/220 of EBV (EBVgp350/220-F VLP) were established and vaccination of mice with EBVgp350/220-F VLPs elicited long-lasting gp350/220-specific antibodies capable of neutralizing EBV *in vitro* (*Ogembo et al., 2015*). In addition, antibodies detected in the sera of immunized animals were predominantly of IgG1 subclass, indicating a Th2-skewed immune response (*Ogembo et al., 2015*). EBV VLPs were also previously reported as a potential prophylactic vaccine candidate against EBV. Immunization of EBV VLPs in mice elicited not only humoral immune response but also cellular mediated immune response (*Ruiss et al., 2011*). CTL immune response characterized by activation of EBV-specific CD8+ T cells induced by EBV VLP is speculated to be attributed to the EBV mRNA encapsidated within the VLP which is then translated in the infected cells and presented via the MHC class I molecules (*Jochum et al., 2012*). However, no viral DNA was detected in the EBV VLPs indicating the particles are non-infectious and non-replicative VLPs (*Ruiss et al., 2011*).

## EXTRACELLULAR VESICLES VERSUS VLPs IN CANCER VACCINE DEVELOPMENT

Extracellular vesicles (EVs) refer to microvesicles and exosomes released by cells into their extracellular space (*Gould & Raposo, 2013*). For decades, EVs have been recognized as cellular garbage with no significant biological function but recently, they have been identified as an important carriers in intercellular signaling, drug delivery and vaccine development (*Nolte-'t Hoen et al., 2016*; *Tominaga, Yoshioka & Ochiya, 2015*). Morphologically, EVs resemble viral particles but cannot replicate as viruses and often encapsulate fragments of proteins, peptides and nucleic acids (*Kourembanas, 2015*; *Wurdinger et al., 2012*). Intriguingly, EVs released from virus infected and tumor cells may also contain viral and tumor antigens, respectively (*Al-Nedawi et al., 2008*; *Nolte-'t Hoen et al., 2016*). For example, tumor cell-derived exosomes were demonstrated to contain a tumor antigen which could effectively prime the CTL immune responses, but the EVs were shown to induce cancerous phenotypic changes in healthy cells, posing a serious risk of tumorigenesis (*Melo et al., 2014*; *Wolfers et al., 2001*). APC derived EVs are likely to be more effective and safer candidates to be recruited in cancer vaccine development as they naturally present MHCs, co-stimulatory and adhesion molecules on their surface, readily

facilitating the activation of B and T cell immunities (*Nolte-'t Hoen et al., 2009*; *Raposo et al., 1996*). Conversely, VLPs lack these surface molecules and activation of B and T cell immunities can be achieved via recognition of PAMP of VLPs (*Yan et al., 2015*). Attributed to the viral origin of the VLPs, they are mostly immunogenic even in the absence of an adjuvant whereas most EVs are poorly immunogenic (*Zhu et al., 2017*). Although EVs can also be modified genetically to carry or display specific antigens, surface modification of EVs is mostly achieved by using the protein transfer technology or the surface display technology in which a foreign protein or a peptide is fused with the transmembrane proteins of the EVs via genetic engineering followed by expression in a suitable protein expression system (*Stickney et al., 2016*).

## Potential drawbacks of VLPs in cancer vaccine development

Despite promising results presented by VLPs, there are two major drawbacks of using VLPs as a platform in cancer vaccine development. One of the major concerns is the pre-existing immunity of an immunized individual against the VLPs which function as nanocarriers for epitopes or therapeutic peptides. This phenomenon is known as carrier induced epitopic suppression (CIES), characterized by suppression of antibody responses directed against the antigen conjugated to an immunogenic carrier due to the pre-existing immunity against the carrier (*Herzenberg & Tokuhisa, 1982*). However, the effect of CIES can be mitigated and the peptide specific immune response can be enhanced by increasing the copy number of peptides displayed on the surface of the VLPs, repeated injections or higher doses of peptide conjugated VLP vaccines (*Jegerlehner et al., 2010*; *Kjerrulf et al., 1997*). Alternatively, CIES can be avoided by careful selection of a VLP-based carrier. For instance, VLPs of non-human origin should be selected in the development of vaccines for human use (*McKee et al., 2012*).

Another impediment in VLP-based vaccine development is the cost of production. A vaccine candidate is unlikely to be competitive in the market if its manufacturing process is not scalable due to cost ineffectiveness even if it yields a promising pre-clinical outcome. For instance, current HPV VLP-based vaccine costs approximately US$360 for the full regimen in the United State, a price which is not widely affordable in many developing countries where the CC is actually most prevalence (*Madrid-Marina et al., 2009*; *Wang & Roden, 2013*). Production cost of VLP-based vaccine is heavily reliant on the design of the VLPs and the expression systems used. Bacterial expression systems are widely adopted for VLPs production due to its inexpensive set up, but the system lacks post-translational modification machinery, which often leads to the formation of misfolded protein, reduced protein solubility, low yield or cell death (*Terpe, 2006*). Therefore, majority of the commercialized vaccines are expressed in eukaryotic cells (yeast, insect or mammalian cells) equipped with post-translational modification machinery. Nonetheless, these expression systems often required greater operational costs and this eventually increases the cost of the end products (*Khan, 2013*; *Kushnir, Streatfield & Yusibov, 2012*).

## SUMMARY

Cancers that emerge following infectious diseases are notorious killers worldwide. Vaccination against such cancers is speculated to be the most effective approach in cancer prevention and treatment. VLPs are promising candidates in cancer vaccine development due to their characteristics such as stability and capability of self-assembly. These prerequisite features are required for packaging and delivery of bioactive materials, such as tumor antigens, antibody fragments, immunodominant regions, short peptides, DNA, or RNA. Construction of chimeric VLPs depends on the nature of the VLPs. Although both non-enveloped and enveloped VLPs can be produced by genetic manipulation, the surface chemical modification and protein transfer technology represent alternatives for the production of non-enveloped and enveloped VLPs, respectively with minimum genetic modifications. In addition, VLPs containing PAMP can be used in the development of self-adjuvating vaccines, mitigating the reliance on the use of adjuvants which is often associated with some undesired side effects.

In treating or preventing HBV-related diseases, VLPs carrying fragments of antibodies against HBsAg and the '*a*' determinant of HBV were invented as new therapeutic and preventive vaccine candidates, respectively (*Pleckaityte et al., 2015*; *Tan & Ho, 2014*; *Yong et al., 2015a*). Furthermore, VLPs carrying HCV glycoproteins E1 and E2 can be potentially developed into HCV preventive vaccines. These chimeric VLPs have been demonstrated to induce specific humoral immune response in immunized mice. With respect to HPV related diseases, approximately 70% of cervical cancer cases caused by the most virulent strains HPV-16 and 18 can be prevented by current prophylactic vaccines on the market (*Nour, 2009*). To combat the remaining 30% of cervical cancer cases caused by less prevalent but, nonetheless, high-risk virulent strains, preparations that utilize highly conserved L2 minor capsid protein of HPV are currently being tested. In particular, a new fusion vaccine has been created by incorporating L2 minor capsid protein of HPV into VLPs. *Schellenbacher et al. (2013)* showed that such multivalent HPV vaccines (L1/L2 chimera) could cross-neutralize more than 20 HPV subtypes both *in vitro* and *in vivo*, paving way to the development of low cost, broad spectrum HPV prophylactic vaccines. On the other hand, the VLP-based therapeutic CC vaccines were developed to target E7 and E6 proteins expressed by the HPV-associated CC. These VLP-based vaccines could represent a curative treatment to patients with pre-existing HPV-associated CC and they can also be engineered to induce a desired immune response by fusing with specific immunostimulatory molecules via genetic engineering approaches. For instance, MHC class II restricted T cells epitope was incorporated into the RHDV VLPs to favor the proliferation and activation of T cells which in turn enhance CTL immune responses (*Jemon et al., 2013*).

Besides oncovirus-related cancers, other forms of cancer without a known etiology agent can also potentially be treated with VLP-based therapeutic vaccines by inserting tumor antigens into the viral envelop or capsid proteins followed by antigen presentation on the target tumor side. Highly aggressive pancreatic cancer may be potentially treated with Trop2 TAA incorporated into the envelope of SIV VLPs, although co-administration with gemcitabine may be required for optimal protection (*Cubas et al., 2011*). Similarly,

the insertion of mesothelin into SHIV VLPs also induced significant anti-tumor responses. Furthermore, prostate cancer may be potentially prevented with immunotherapy that comprises PSA incorporated MPy VLPs. Co-administration of VLP-based vaccine loaded into DCs with CpG oligonucleotides as adjuvant could protect against tumor outgrowth. In preparations of putative vaccines against breast cancer, the TAA Her2 was incorporated into murine polyomavirus VLPs (*Tegerstedt et al., 2005*) and enveloped influenza VLPs (*Patel et al., 2015b*). Both approaches led to the induction of cellular and humoral immune responses; moreover, loading VLPs into DCs was demonstrated to synergistically improve immunization efficiency in a murine polyomavirus model. VLPs utilized in therapeutic skin cancer vaccine preparations include bacteriophage Qβ, HBc capsid, murine polyomavirus, and RHDV. Potential VLP-based therapeutic and preventive cancer vaccines have also been proposed for lung and EBV related cancers.

Morphologically, EVs resemble viral particles, and the former are important carriers in intercellular signaling, vaccines and drug delivery systems. Similar to VLPs, EVs were employed as a platform for cancer vaccine development owing to their unique features in harboring and displaying tumor antigens. Nevertheless, the challenges faced by EV-based carriers include safety, immunogenicity and efficacy, which must be addressed properly in the future. In fact, due to the natural occurrence of EVs in the cells, most of the studies about EV-based carriers in the past were mainly focused on drug delivery system instead of vaccine development (*Ohno, Drummen & Kuroda, 2016*; *Tominaga, Yoshioka & Ochiya, 2015*). However, EVs derived from the surface of APC containing MHCs, co-stimulatory and adhesion molecules have potential in vaccine development.

Pre-existing immunity of the immunized individuals and cost of vaccine production are the two major drawbacks of using VLPs as a platform in vaccine development. Pre-existing immunity against VLPs could interfere the efficacy of any subsequent vaccination involving the same VLP via CIES. Although the effect of CIES can be alleviated by several approaches, a careful selection of VLPs could avoid CIES. On the other hand, vaccine production must be cost effective and scalable. Ideally, to eradicate a preventable disease, a vaccine must be affordable for everyone in the globe, especially the developing countries where the diseases are actually most prevalent.

In conclusion, efficacy of the VLP-based vaccine is highly dependent on the selection of tumor antigens and the design of VLPs. In order to produce an effective VLP-based vaccine, the genetic factor of an individual tumor has to be taken into consideration during the selection of the tumor antigen because mutations of the tumor cells vary considerably from person to person even they are diagnosed with the same kind of cancer (*Castiblanco & Anaya, 2015*; *Ott et al., 2017*). In the process of selecting and designing VLPs, the effect of CIES due to pre-existing immunity must be mitigated or possibly avoided. VLPs have to be designed in a way that they could effectively activate the innate and adaptive immunities and this could be achieved by selecting highly immunogenic VLPs, conjugation or co-administration of VLPs with adjuvants or other immunostimulatory molecules. All in all, VLPs have great potential in cancer vaccine development.

### Funding

Hui Kian Ong was financially supported by the Graduate Research Fellowship (GRF) from Universiti Putra Malaysia (UPM) and the MyBrain Scholarship from the Ministry of Higher Education, Malaysia. This work was supported by the Fundamental Research Grant Scheme (FRGS; Grant number 04-02-13-1323FR) of the Ministry of Higher Education, Malaysia and Research University Grant (Grant number: GP-IPS/2016/9505700) of Universiti Putra Malaysia. The funders had no role in study design, data collection and analysis, decision to publish, or preparation of the manuscript.

### Grant Disclosures

The following grant information was disclosed by the authors:
Graduate Research Fellowship (GRF).
MyBrain Scholarship.
Fundamental Research Grant Scheme (FRGS) of the Ministry of Higher Education, Malaysia: 04-02-13-1323FR.
Research Grant of Universiti Putra Malaysia: GP-IPS/2016/9505700.

### Competing Interests

The authors declare there are no competing interests.

### Author Contributions

- Hui Kian Ong planned the contents of the manuscript, performed data searches, analyzed the data, wrote the paper, prepared figures and/or tables.
- Wen Siang Tan contributed analysis tools, reviewed drafts of the paper.
- Kok Lian Ho planned the contents of the manuscript, analyzed the data, wrote the paper, supervision.

### Data Availability

Raw data was not generated for this study; this paper is a literature review.

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
