# Peer review of "Virus like particles as a platform for cancer vaccine development"

_PeerJ, doi:10.7717/peerj.4053_

## Round 0.1 · original submission · Major Revisions

I have now received comments from 4 reviewers. Please revise your manuscript according to the reviewers' comments and address each concern fully. I look forward to receiving your revised version.

Reviewer 1 ·

Basic reporting

The summary highlights the importance and qualities of VLP's, however, I suggest improving the wording on lines 23-28 and creating a single paragraph, to avoid repetition of "VLP's" in each sentence.

In line 88, I suggest that more information about the VLPs should be included in the text, for example: classification, synthesis method, architecture (wrapped or not wrapped) or chimeras. This will give a greater impact to the review.

The revision would benefit from the inclusion of a figure where the production of VLPs is schematized, for example as shown by Antonina Naskalaska et al. (Polish Journal of Microbiology 2015, Vol. 64, No 1, 3-13), Dan Yan et al. (Appl Microbiol Biotechnol (2015) 99: 10415-10432) by mention some authors.

Experimental design

Does not apply

Validity of the findings

Does not apply

Additional comments

- Has the field been reviewed recently? If so, is there a good reason for this review (different point of view, accessible to a different audience, etc.)?
I consider the review to be of great interest since it exposes the use of VLPs as a prophylactic platform for the prevention and treatment of various types of cancer of importance in the world. The last review that I could find about VLP's as a vaccination strategy is 2014, so I consider that this work brings an update in this field so nove.

- Does the Introduction adequately introduce the subject and make it clear who the audience is/what the motivation is?
The summary highlights the importance and qualities of VLP's, however, I suggest improving the wording on lines 23-28 and creating a single paragraph, to avoid repetition of "VLP's" in each sentence.
I made this recommendation because I consider that from the first moment that the VLPs are mentioned it is clear that the information that will be read in the text will be referring to those particles so it is not necessary to repeat VLP's at the beginning of the sentences in lines 23- 28.

In line 88, I suggest that more information about the VLPs should be included in the text, for example: classification, synthesis method, architecture (wrapped or not wrapped) or chimeras. This will give a greater impact to the review.
In line 88, where the author begins to describe the VLP's, I suggested that the author provide more information on the origin of the VLPs because in the introduction little is said about these particles and their classification, although the revision is not focused on obtaining VLPs, I consider that to provide general information about the various types of VLP's will provide more information to readers

The revision would benefit from the inclusion of a figure where the production of VLPs is schematized, for example as shown by Antonina Naskalaska et al. (Polish Journal of Microbiology 2015, Vol. 64, No 1, 3-13), Dan Yan et al. (Appl Microbiol Biotechnol (2015) 99: 10415-10432) by mention some authors.
I cited these authors as an example because in their publications they showed schemes that explain very clearly how the VLPs act. I believe that these works can provide the author with a different and clearer perspective on how to outline the role of VLPs as a prophylactic or vaccine platform against cancer. But if the editor thinks the scheme is appropriate, please ignore my recommendation.

- Is the Survey Methodology consistent with a comprehensive, unbiased coverage of the subject. If not, what is missing?
The research platforms used to obtain the information allow to review several areas of research carried out with these VLPs, which is why I consider it to be an adequate methodology for this manuscript.

- Are sources adequately cited? Quoted or paraphrased as appropriate.
All references are properly quoted.

- Is the review organized logically into coherent paragraphs/subsections?
Yes

Reviewer 2 ·

Basic reporting

The authors performed an excelente review of several paper about the potential use of VLPs as cancer therapeutic molecules. The review is well redacted and profound.

Minor changes.
Authors mention that VLPs are promising molecules to fight cancer, however, they must include a paragraph with the potential drawback of this approach. For example, several reports have showed that preexisting human immunity against several VLPs reduced the potential use of this particles. Authors should declare why only few of them has been successfully enter in clinical trials and less are have been licensed.

Experimental design

no comments

Validity of the findings

no comments

Additional comments

no comments

Reviewer 3 ·

Basic reporting

The review is well written and follows the guidelines of the journal. However, the manuscript has some deficiencies since, according to its title “VLPs as platforms...”, it should address the issue from two points of view: VLPs against cancer-causing viral infections (preventive) and VLPs developed against cancer (therapeutics). Nevertheless, most of the content is only focused on VLPs as preventive vaccines against cancer-associated viral infections such as those related to HBV, HCV, HPV and HHV-4 (EBV). It is also important to consider that these oncogenic viruses are associated to malignant processes, thus VLPs vaccines are developed to prevent viral infection that in turn has an impact on the incidence of cancer.
VLPs as therapeutic vaccines are only addressed in the section 2.3, about pancreatic, prostate, breast, skin and lung cancer. In the case of HCC, the information in that section is mixed.
On the other hand, authors do not discuss all the strategies that are being used for development of VLPs as a carrier of antigens, but only show in Figure 1, one of two main strategies for generation of VLPs (genetic fusion). It would be important to contrast at least with a second strategy; authors might consider Eriksson et al., 2011 to extend information. Another important point that the authors do not address is the viral origin of the VLPs, since origin is important from the point of view of the immune response that could be presented
Therefore, authors should consider dividing the review into two topics: preventive VLPs and therapeutic VLPs or should focus the review only on therapeutic VLPs.

Experimental design

No comment

Validity of the findings

No comment

Additional comments

Minor revision
• In line 119 the word ("Table 2") would be better if it said ("see Table 2")
• In line 207, the drug is Gardasil not Gadrasil.
• The paragraph on lines 431-433 is confusing or repeated
Major revision
• In lines 105-107, compared to that, do VLPs represent “a much lower risk of autoimmunity”? Which vaccines have autoimmunity problems or do they exhibit auto-antigens?
• In line 150 quotes the work of Pleckaityte et al. 2015, and it is mentioned that the VLP developed by this group has therapeutic potential, but in the work show in vitro results of viral neutralization. Can the vaccine cure hepatocellular cancer or only prevent HBV infection? Or control and eliminate chronic HBV infection in people but without hepatocellular cancer?
• The paragraph of line 197 to 200, is a conclusion of the work of Ding et al. 2009? Or it is a contribution of the work of Su et al. 1998, if it is the latter, then that work should be better described.
• The paragraph of lines 250-251, refers to the VLPs developed by Martin Caballero et al. 2012 and Jemon et al. 2013, these proved to be therapeutic, then, according to the paragraph would be prophylactic or therapeutic?
• All section 3.7.2 EBV should go after section 3.2 HPV.
• The finding of mRNA in the EBV VLP in the work of Russ et al. 2011 could be more discussed (Line 436).
• Section 4.0 (Summary), needs more discussion about the different ways to obtain preventive and therapeutic chimeric VLPs. The works of Martin Caballero et al. 2012 and Jemon et al. 2013, can be used to extend the discussion. On the other hand, the conclusion is very general, because some VLPs have been promising, but many others have not.
In the figure it is not clear if what follows after the mouse are more viruses or are dendritic cells?

Reviewer 4 ·

Basic reporting

This is a very interesting, original, and well-written review by Ong et al., regarding virus like particles as platforms for cancer vaccine development. The authors presented almost all existing literature in an innovative and entertaining form. However, I would like to see some more recent literature which describes extracellular vesicles and their close resemblance with VLPs. I would like to see authors critically reflect on how VLPs are different than exosomes/microvesicles and that role can these extracellular vesicles play in vaccine development platform ?

Please see articles such as
1) Extracellular vesicles and viruses: Are they close relatives? Nolte-'t Hoen E, Cremer T, Gallo RC, Margolis LB. Proc Natl Acad Sci U S A. 2016 Aug 16;113(33):9155-61. doi: 10.1073/pnas.1605146113. Epub 2016 Jul 18.
2) Microvesicles and Vesicles of Multivesicular Bodies Versus “Virus-Like” Particles. Albert J. Dalton. JNCI: Journal of the National Cancer Institute, Volume 54, Issue 5, 1 May 1975, Pages 1137–1148, https://doi.org/10.1093/jnci/54.5.1137
3) Extracellular Vesicles and Their Convergence with Viral Pathways. Thomas Wurdinger,1,2 NaTosha N. Gatson,3 Leonora Balaj,1 Balveen Kaur,3 Xandra O. Breakefield,1 and D. Michiel Pegtel4. Advances in Virology. Volume 2012 (2012), Article ID 767694, http://dx.doi.org/10.1155/2012/767694

Experimental design

Original content within the scope of the journal

Validity of the findings

Conclusions are well-stated, and data has been presented in a very entertaining manner,

---

## Round 0.2 · accepted · Accept

I appreciate very much your modifying this manuscript, and leave one suggestion from reviewer 3 regarding a slight change in figure 2, which is up to you to follow or not. Congratulations again and I look forward to handling future submissions to PeerJ from you.

Reviewer 1 ·

Basic reporting

No comment

Experimental design

No comment

Validity of the findings

No comment

Additional comments

The modifications made along with the information that was added and the figures positively increases the academic contribution of this article

Reviewer 2 ·

Basic reporting

Without comments

Experimental design

Without comments

Validity of the findings

Without comments

Additional comments

Without comments

Reviewer 3 ·

Basic reporting

In Figure 2, I think it would be best if the self-assembly was first and then the release and budding to continue with the VLPs

Experimental design

no comment

Validity of the findings

no comment